# *Clostridioides difficile* Infection in Patients after Organ Transplantation—A Narrative Overview

**DOI:** 10.3390/jcm11154365

**Published:** 2022-07-27

**Authors:** Sylwia Dudzicz-Gojowy, Andrzej Więcek, Marcin Adamczak

**Affiliations:** Department of Nephrology, Transplantation and Internal Medicine, Medical University of Silesia, Francuska 20-24, 40-027 Katowice, Poland; sdudzicz@sum.edu.pl (S.D.-G.); awiecek@sum.edu.pl (A.W.)

**Keywords:** *Clostridioides difficile* infection, solid-organ transplantation, hematopoietic stem cell transplantation

## Abstract

*Clostridioides difficile* infection (CDI) is one of the most common causes of antibiotic-associated diarrhea. The pathogenesis of this infection participates in the unstable colonization of the intestines with the physiological microbiota. Solid-organ-transplant (SOT) patients and patients after hematopoietic stem cell transplantation are more prone to CDI compared to the general population. The main CDI risk factors in these patients are immunosuppressive therapy and frequent antibiotic use leading to dysbiosis. The current review article provides information about the risk factors, incidence and course of CDI in patients after liver, kidney, heart and lung transplantation and hematopoietic stem cell transplantation.

## 1. Methodology

This literature review was based on the PubMed database using the following keywords: *Clostridium difficile* infection, *Clostridioides difficile* infection, microbiota, dysbiosis, solid-organ transplantation, hematopoietic stem cell transplantation, kidney transplantation, liver transplantation, heart transplantation and lung transplantation. Original articles, review articles and case reports from 2001 to 2022 were included in the analysis.

## 2. Microbiota

The intestinal microbiota consist of microorganisms that live in the human gastrointestinal tract, such as various species of bacteria, viruses, fungi and protozoa. The intestinal microbiota composition depends on several factors, such as diet, exposure to environmental microbiota or toxins, as well as on occurring diseases and used drugs, including antibacterial agents [1,2]. The physiological role of the intestinal microbiota is complex. It is to maintain the intestinal homeostasis and integrity of the gastrointestinal epithelium, regulate the immune system’s function, prevent colonization and infection with pathogenic microbes and provide nutritional benefits to the human host [3]. Dysbiosis is an imbalance in the composition and function of these microorganisms. It is manifested by a reduction in the number of beneficial bacteria, an increase in the number of potentially pathogenic bacteria and a loss of microbiota diversity and richness [4]. 

Organ transplantation recipients often show a reduction in the microbiome’s diversity and an increase in potentially pathogenic bacteria, such as *Proteobacteria.* In addition, it is associated with a decrease in the production of short-chain fatty acids (SCFA), which have a positive effect on the functioning of the intestinal epithelial barrier, modulate the immune response and prevent the proliferation of cancer cells. It is unclear whether this is due to factors associated with organ transplantation (e.g., immunosuppressive therapy, more frequent antibiotic therapy and hospitalizations) or whether it is due to baseline organ dysfunction. The characteristics of dysbiosis for each type of organ transplant are discussed in the paragraphs below [5].

The analysis of the microbiota composition is completed using 16S rRNA sequence analysis. The choice of 16S rRNA was determined, among others, by common occurrence in bacteria and its highly conservative nature. It consists of isolating genomic DNA from the tested material, on the template of which, in the PCR reaction, the fragment encoding the 16S rRNA subunit is amplified. Identification is based on the search for homology between the sequence of the product obtained and the sequences stored in available databases. Strains showing at least 95% sequence identity of the 16S rDNA are classified into the same genus, while 97% or greater similarity determines the species affiliation of the strain [6].

### 2.1. Clostridioides difficile Infection

*Clostridioides difficile* is a strictly anaerobic Gram-positive, rod-shaped bacteria, with the ability to produce spores. Spores of this bacteria show increased resistance to high temperature and are resistant to acids and antibacterial drugs. Transmission of the infection occurs via the fecal–oral route. After ingestion and transfer into the duodenal lumen, spores stimulated by the contact with bile acids transform into vegetative forms. 

*Clostridioides difficile* infection is one of the most common causes of antibiotic-associated diarrhea [7]. In recent years, there has been an increase in the incidence of CDI and a more frequent occurrence of severe and complicated forms of this infection. Moreover, recently, overall mortality in CDI increased from 4.5% to 5.7% [8]. This is due to the worldwide spread of the NAP1/BI/027 *Clostridioides difficile* strain, characterized by increased synthesis of toxins A and B, production of the binary toxin, greater capacity to form spores and greater resistance to fluoroquinolones [8,9]. The main factor determining the pathogenicity of *Clostridioides difficile* is the production of A, B and binary toxins. Damage to the intestinal mucosa and excess mucus production results in diarrhea, colitis and the formation of pseudo-membranes composed of inflammatory cells, fibrous exudates and necrosis. 

### 2.2. Clostridioides difficile Infection in Patients after Organ Transplantation

Solid-organ transplant (SOT) patients are more prone to CDI compared to the general population. The main risk factors of *Clostridioides difficile* infection in these patients are immunosuppressive therapy and frequent antibiotic therapy used for prophylactic or therapeutic purposes [10]. Exposure to toxigenic strains of *Clostridioides difficile* may lead, in SOT patients, similarly to the general population, to asymptomatic colonization of the large intestine or symptomatic CDI. Persistent intestinal colonization by *Clostridioides difficile* is found in 4–15% of the population and most people have demonstrated periodic colonization [11]. Clinically, CDI may present as: diarrhea of varying severity, colitis without pseudomembranes, pseudomembranous colitis, fulminant colitis with megacolon toxicum, paralytic obstruction or perforation of the colon, sepsis or multi-organ failure. CDI’s common clinical symptoms include watery diarrhea, lower abdominal pain, fever, nausea, vomiting and malaise. From 15% to 25% of patients with CDI develop a relapse after treatment [12]. In the SOT patients, the risk of recurrence is approximately 20%, but in some of them (e.g., patients after lung or heart transplantation), it may increase up to 33%. The most common cause of this condition is the unstable colonization of the intestines with the physiological microbiota, less often the failure to produce IgG antibodies against toxins A and B or resistance of *Clostridioides difficile* to the antibacterial drugs used, i.e., vancomycin or fidaxomicin [13].

#### 2.2.1. *Clostridioides difficile* Infection Diagnosis

The diagnostic procedures for CDI in SOT patients are similar to in the general population. The current guidelines for CDI diagnosis were published by the *Infectious Diseases Society of America* (IDSA) and the *Society for Healthcare Epidemiology of America* (SHEA) in 2017. *Clostridioides difficile* infections are diagnosed in patients with diarrhea or *megacolon toxicum* and with the occurrence of one of the following criteria: the presence of toxins A and/or B in the stool or a toxin-producing strain of *Clostridioides difficile* in the stool culture or other diagnostic methods or evidence of pseudomembranous enteritis on endoscopic examination, during surgery or on histopathological examination. The diagnostic material is a stool sample taken from a patient suspected of CDI. In the absence of diarrhea in the case of bowel obstruction, it is necessary to collect a rectal swab for molecular testing or culture for the presence of a toxin-causing *Clostridioides difficile* strain. In diagnostics, a multi-step algorithm is used. In the first stage, it is recommended to complete a highly sensitive test-nucleic acid amplification test (NAAT) or enzyme immunoassay to detect glutamate dehydrogenase (EIA GDH). In the case of a positive result in the second stage, enzyme immunoassays are used to detect toxins A and B. A positive test result confirms CDI. In the case of a negative result, it is advisable to consider the presence of CDI based on the patient’s clinical status assessment or complete NAAT test (in the case of GDH EIA in the first stage) or stool culture for *Clostridioides difficile* with the determination of bacterial toxigenicity [14].

#### 2.2.2. Prophylaxis of *Clostridioides difficile* Infection

Due to the increased incidence of CDI and its severe forms and relapses in patients after solid-organ transplantation and hematopoietic stem cell transplantation, it is important to use effective prophylaxis in these patients. It is important to use non-specific prophylaxis to limit the transfer of infection, such as: using contact precautions, such as disposable gloves and an apron, frequent handwashing with soap and water, environmental disinfection and, in a case of diagnosed CDI, isolation of the patient [14,15,16]. 

Taking into account that one of the causative factors of CDI is dysbiosis, it is also recommended in the general population, to administer probiotics during antibiotic therapy to prevent the disruption of the intestinal microbiota. Limited data are available on the safety and efficacy of probiotics used in patients after organ transplantation. Dudzicz et al., in a retrospective study, observed a significant decrease in the CDI incidence rate (from 44.9 to 7.2 per 1000 patients hospitalized; *p* = 0.005) after implementation of a *Lactobacillus plantarum 299v* (LP299v) strain as CDI prevention in patients during immunosuppression and antibiotic therapy and significantly increased the CDI incidence (from 7.2 to 34.0 per 1000 hospitalized patients; *p* = 0.025) after cessation of this method of prophylaxis [17]. Grąt et al. analyzed the effectiveness of probiotics in infection prevention (including CDI) in 55 patients in the early postoperative period after liver transplantation. In patients in such an early period after successful liver transplantation receiving a combined probiotic preparation containing *Bifidobacterium bifidum*, *Lactobacillus acidophilus*, *Lactobacillus casei* and *Lactococcus lactis*, a significantly lower incidence of infections compared to patients who received placebo was found (30-day infection rates: 4.8% versus 34.8%, *p* = 0.02; 90-day infection rates: 4.8% versus 47.8%, *p* = 0.002). Moreover, in patients receiving probiotics, a faster improvement in the biochemical parameters of the transplanted liver function was observed: lower plasma bilirubin concentration and a faster decrease in the activity of aspartate and alanine aminotransferases in plasma were found (*p* = 0.02, *p* = 0.03 and *p* = 0.03, respectively) [18]. Rayes et al., in a prospective, randomized, placebo-controlled study, analyzed the incidence of postoperative infections in 95 patients after liver transplantation. Patients were divided into three groups, differing in the method of early enteral nutrition. Significantly more infections occurred in patients who received preventive antibacterial treatment for the so-called selective bowel decontamination (SBD) compared to the LP299v and oat fiber group. The incidence of infections in the placebo group (heat-inactivated LP299v and oat fiber) was also lower than in patients with SBD [19].

Decreasing unnecessary use of gastric-acid-suppressant medications (proton pump inhibitors (PPIs) and histamine H2 receptor blockers) only to clear clinical indications may also reduce the incidence and recurrence of CDI. In a meta-analysis of 16 studies that included 7703 subjects, Tariq et al. found a higher incidence of recurrent CDI in patients with gastric-acid-production suppression (22.1% compared with 17.3% in patients without gastric acid suppression, OR 1.52; 95% CI: 1.20–1.94; *p* < 0.001) [20]. In another meta-analysis of 50 studies, including 342,532 subjects, Cao et al. found significant association between PPI use and CDI risk (OR: 1.26; 95% CI: 1.12–1.39). Increasing the pH of gastric fluid due to the use of proton pump inhibitors facilitates the growth of potentially pathogenic bacterial strains and fungi in the gastrointestinal tract by dysbiosis induction. In addition, reducing the acidity in the stomach environment may enable or facilitate conversion from spores to vegetative forms of *Clostridioides difficile* in the upper gastrointestinal tract [21].

#### 2.2.3. Treatment of *Clostridioides difficile* Infection

CDI treatment in patients after solid-organ transplantation and hematopoietic stem cell transplantation is the same as in the general population. CDI treatment is carried out in accordance with the current guidelines of the *European Society of Clinical Microbiology and Infectious Diseases* (ESCMID) published in 2021 [22]. The main drugs used in the treatment of CDI are vancomycin and fidaxomicin. 

Orally administered vancomycin is only slightly absorbed from the gastrointestinal tract and, therefore, reaches high concentrations in the intestinal lumen. Clinical trials showed high effectiveness of CDI treatment with vancomycin—it was 98% in mild forms of CDI and 97% in severe forms of CDI. However, its use is associated with the risk of vancomycin-resistant enterococci (VRE) selection and, due to the broad spectrum of activity, with the risk of dysbiosis, which increases the risk of recurrent CDI and *Clostridioides difficile* colonization. 

Fidaxomicin is a macrolide antibiotic with a narrow spectrum of activity against aerobic and anaerobic Gram-positive bacteria. After oral administration, it is not absorbed from the gastrointestinal tract. Its efficacy in CDI treatment is comparable to that of vancomycin—88% efficacy in patients treated with fidaxomicin compared to 86% in patients treated with vancomycin. Moreover, no adverse effect of fidaxomicin on the physiological intestinal microbiota was observed. Its advantage is a lower risk of relapse compared to patients treated with vancomycin (15% vs. 25%) due to the prevention of the formation of spores of *Clostridioides difficile* [23,24]. Different treatment algorithms are recommended, depending on the severity of CDI or the incidence of relapse, as presented in Figure 1.

One of the effective treatments for recurrent CDI is the transfer of the intestinal microbiota (FMT). This method of therapy may be considered in recurrent CDI, especially in the event of failure of pharmacological treatment. There are few published studies of FMT in patients undergoing immunosuppressive therapy. Friedman-Moraco et al. described cases of two patients after organ transplantation (after kidney and lung transplantation) with recurrent CDI, in whom FMT treatment proved to be safe and effective [25]. In a study by Lin et al., involving five patients after organ transplant (four patients after kidney transplantation and one patient after pancreatic transplantation) with recurrent CDI, in four out of five patients, there was no recurrence of infection after one FMT. The most common adverse reaction to FMT in these patients was cramp-like abdominal pain and constipation [26]. Webb et al. assessed the efficacy and safety of FMT in a group of patients after hematopoietic cell transplantation. Six of seven (i.e., 85.7%) patients had no recurrence; one patient recurred at day 156 post FMT after therapy with an antibiotic given orally and required repeat FMT, after which no further recurrence was observed. In this study, no serious adverse events were noted and all-cause mortality was 0% [27].

Bezlotoxumab is a human monoclonal antibody against *Clostridioides difficile* toxin B that binds and neutralizes the above-mentioned toxin. Currently, bezlotoxumab is approved to prevent recurrent CDI in high-risk adult patients. Its efficacy has been confirmed in large, randomized, double-blind trials: MODIFY I and MODIFY II. Bezlotoxumab is administered as a single intravenous dose during oral antibiotic therapy in patients with CDI with a high risk of CDI recurrence [28]. Gerding et al., in a randomized clinical trial with placebo in a subpopulation of patients with risk factors for CDI recurrence, including immunosuppressive therapy, found a significant reduction in the risk of CDI recurrence in the bezlotoxumab group compared to the placebo group (28.1 vs. 54.3%, respectively) [29]. The effectiveness of bezlotoxumab in SOT patients in reducing the frequency of CDI recurrences was also analyzed by Kerr et al., in a retrospective study. The above-mentioned study included 21 SOT patients (12 treated with bezlotoxumab, 9 received placebo) with CDI. All participants received standard antibiotic therapy. Kerr et al. found a reduction in CDI recurrence from 33% to 9% compared to controls during the 90-day follow-up period [30].

Another interesting recent report described the use of an oral therapeutic microbiome preparation in the treatment of CDI relapses. This microbiome preparation, called SER-109, was composed of live, purified spores from the *Firmicutes* bacteria. Feuerstadt et al., in a phase 3, double-blind, randomized, placebo-controlled trial, with 182 patients from the general population, observed that the occurrence of CDI recurrence was 12% in the SER-109 group and 40% in the placebo group (RR 0.32; 95% CI, 0.18 to 0.58; *p* < 0.001). The most common adverse effect of the studied drug was mild to moderate gastrointestinal complaints. The effectiveness of the *Firmicutes* strain used is most likely due to competition with *Clostridioides difficile* for essential nutrients or modulating bile acid profiles to restore colonization resistance. In the future, SER-109 therapy may be a valuable alternative to FMT, which may carry the risk of transmitting undetected pathogens that may lead to a deterioration in the patient’s general health status. SER-109 may also be used not only for the treatment of recurrent CDI but also other diseases, with a pathogenesis that may be related to dysbiosis [31]. This method of CDI prophylaxis undoubtedly needs study in patients after solid-organ transplantation and hematopoietic stem cell transplantation.

## 3. Liver Transplantation

Intestinal dysbiosis is one of the disorders reported in patients with liver cirrhosis, especially in patients with decompensated cirrhosis and hepatic encephalopathy. The CDI risk in patients with liver cirrhosis ranges from 0.9% to 5.7% in the available published studies [32,33]. Dysbiosis in patients with liver cirrhosis may be caused by dysfunction in bile acid metabolism and production, hyperammonemia and, as a result, lead to bacterial translocation due to increased permeability of the intestinal barrier [34]. Clinical studies describe an improvement in intestinal microbiota function after liver transplantation. Bajaj et al. analyzed data on the microbiota composition in 45 patients after liver transplantation. Based on the analysis of microbiota in stool samples, an increase in bacterial richness of microbiota in patients after liver transplantation was found (index Chao1—nonparametric method for estimating the number of species in a community, pre-LT patients 468.9 ± 263.1 vs. post-LT patients 694.9 ± 240.6; *p* < 0.05). It was also found that, in liver transplant recipients, whose cognitive functions did not improve after transplantation, the number of *Proteobacteria* and *Enterobacteriaceae* increased significantly and the number of commensal bacteria *Firmicutes* decreased (median in subjects before liver transplant compared to patients after liver transplantation 0% vs. 12%; *p* = 0.04) [35]. In another study, Bajaj et al., including 40 liver transplant recipients, showed an increase in diversity of the intestinal microbiota after liver transplantation compared to the pre-transplant period. Control microbiota evaluation was completed from 4 to 10 months after liver transplantation (Shannon microbial diversity index—measure of diversity that combines the number of species in a given area and their relative abundances, 2.1 ± 0.7 vs. 1.6 ± 0.7; *p* = 0.001). In the post-transplant period, a decrease in the number of pathogenic bacterial species, such as *Escherichia*, *Shigella* and *Salmonella,* and an increasing number of commensal bacteria, such as *Lachnospiraceae* and *Ruminococcaceae,* were also found [36]. In addition to the improvement in intestinal dysbiosis after successful liver transplantation, Wijarnpreecha et al., in a retrospective cohort study, included 1665 liver transplanted patients with CDI and observed a higher incidence of CDI in patients after liver transplantation compared to the general population (OR 2.78; 95% CI: 2.44–3.16). Liver transplant patients diagnosed with CDI had a higher risk of shock, acute kidney injury (AKI), ICU hospitalization and organ failure compared to non-CDI patients [37]. In a retrospective analysis of 192 liver transplant patients, Sullivan et al. found that patients with a higher *Model for End-Stage Liver Disease* (MELD score) before liver transplantation had a higher risk of developing CDI (mean, 24.4 vs. 19.8; *p =* 0.04). Moreover, CDI was more common in patients with nonalcoholic steatohepatitis (18% vs. 5%, *p =* 0.031) and coinfection with hepatitis C virus and human immunodeficiency virus (11% vs. 1%, *p =* 0.021). Additionally, in a multivariate analysis, receiving an organ from a living donor and a MELD score of 20 or more was associated with a significantly higher CDI risk (HR 3.769, *p* = 0.0058 and HR 2.897, *p* = 0.010, respectively) [38].

## 4. Kidney Transplantation

In patients with chronic kidney disease, significant changes in the intestinal microbiota have been observed, including increasing the number of *Firmicutes*, *Actinobacteria* and *Proteobacteria* and reducing the number of *Bifidobacteria* and *Lactobacillus*. The most common causes of dysbiosis in patients with chronic kidney disease are low-dietary-fiber supply, slow down in the passage of the intestinal tract and excessive multiplication of bacteria in the intestinal lumen. Due to the increased concentration of urea, uric acid and hydrogen ions in the plasma, these substances are also secreted in greater amounts into the lumen of the gastrointestinal tract, which creates environmental conditions favorable for the multiplication of pathogenic bacteria [39]. According to frequent antibiotic exposure and frequent hospitalizations, impaired immune system, older age of the patients and the above-mentioned causes of dysbiosis, patients with CKD are especially vulnerable to CDI development [40]. In a meta-analysis of 20 case-control, cohort and cross-sectional studies by Phatharacharukul et al., the risk of CDI in CKD patients was higher than in the general population (RR 2.63, 95% CI: 2.04–3.38) [41]. Keddis et al., based on data from the US National Hospital Discharge Survey of 162 million patients, discharged from hospital in 2005–2009, found CDI in patients with CKD with a frequency of 1.49% compared with 0.70% in patients without CKD [42]. The incidence of CDI in these patients also increases with renal function deterioration and reaches: 2.46% in CKD stages 1 and 2, 21.2% in CKD stages 3 to 5 and 43.5% in CKD stage 5 on renal replacement therapy [42]. After kidney transplantation, disturbances in the composition of the gut microbiota have also been observed [43]. In kidney transplanted patients, it has been found that immunosuppressants significantly change the composition of the intestinal microbiome. One of the underlying mechanisms is diarrhea caused by mycophenolate mofetil. In patients treated with mycophenolate mofetil, the histopathological findings of intestinal mucosa biopsy showed, among others: crypt cell apoptosis, atrophy of the crypt, crypt abscesses with eosinophil infiltrates, focal cryptitis, ulcerations and erosions. These changes can lead to colitis and chronic diarrhea and, as a consequence, disturbances in the composition of the intestinal microbiota. Other causes of diarrhea in kidney transplant recipients may be due to systemic infections occurring in transplant recipients, such as cytomegalovirus (CMV) infection, and due to post-transplant inflammatory bowel disease (IBD), caused by dysregulation of the intestinal immune system through the use of immunosuppressive drugs. In a three-month follow-up study of 71 kidney transplant patients, Lee et al. found a reduction in microbiome diversity in patients with diarrhea compared to the group without diarrhea (Shannon diversity index 2.4 vs. 3.1, *p =* 3 × 10^−7^, Wilcoxon rank-sum test). The results of the above study might suggest the role of dysbiosis in the pathogenesis of diarrhea in these patients [43]. Swarte et al. analyzed the gut microbiome using 16S rRNA sequencing, finding a reduced diversity of the microbiome in patients after kidney transplantation compared to the controls (Shannon diversity index, 3.4 vs. 3.7, *p* < 0.001). The most diminished heterogeneity in the renal transplant patients was in subjects receiving mycophenolate mofetil (*p* < 0.01). Moreover, in patients after kidney transplantation, an increased number of *Proteobacteria* and a decreased presence of *Actinobacteria* and butyrate-producing bacteria was found [44]. 

## 5. Lung Transplantation

Infection is one of the most important complications after lung transplantation. It is the most common cause of death in lung recipients in the first 12 months after transplantation. The risk factors of infection in these patients are: constant contact with the external environment and pathogens, weakened cough reflex and abnormal mucociliary clearance and, as a result, the persistence of potentially pathogenic microorganisms in the respiratory tract caused by graft denervation, and the use high doses of immunosuppressive drugs. To minimize the risk of infection, antibiotic prophylaxis is used. The most frequently used antibiotics are the first- and second-generation cephalosporins. Unfortunately, long-term antibiotic therapy and the use of the above-mentioned antibiotics promote CDI development [45,46]. In a retrospective analysis of 500 patients after lung transplantation, Whiddon et al. found CDI in 6% compared to 1–2% of the general population [47]. In another retrospective analysis, by Lee et al., of 388 patients, 89 recipients (22.9%) developed CDI, of which 27 recipients (7.0%) were hospitalized directly after transplantation. One of the CDI risk factors in these subjects was the length of stay in the hospital in the postoperative period (HR 1.02; 95% CI: 1.01–1.03). In lung transplant recipients who developed CDI, the risk of death was higher (HR 1.61; 95% CI: 1.02–2.52), especially when CDI occurred in the first 6 months after transplantation (HR 1.96; 95% CI: 1.14–3.36) [48]. In a prospective multicenter study by Dubberke et al., in 229 patients, the CDI incidence among lung transplant recipients was 13.1% 1 year after transplantation. There was no significant difference in mortality between CDI cases and controls during the early period after lung transplantation (*p* = 0.52), but mortality was significantly higher among CDI cases during the late period after lung transplantation (HR = 1.80 [95% CI: 1.2–2.8]; *p* = 0.007) compared to controls [49]. Bajrovic et al., in a single-center, retrospective cohort study in a group of 636 patients after lung transplantation, observed a statistically significant 86% reduction in CDI incidence in the vancomycin prophylaxis group (rate ratio 0.14; 95% CI: 0.0068–0.73; *p* = 0.01). Prophylaxis with vancomycin at a dose of 125 mg twice daily by the oral route was started on day 6 after transplantation and continued for 10 days [50].

## 6. Heart Transplantation

In patients after a heart transplantation, the highest incidence of CDI occurs approximately 1 month after transplantation [51]. One of the CDI risk factors in these patients is severe hypogammaglobulinemia. In heart recipients, hypogammaglobulinemia is an independent CDI risk factor (RR 5.8; 95% CI: 1.05 to 32.1; *p* = 0.04). In a study involving 235 heart transplant patients, Muñoz et al. found CDI in 14.9%. The incidence of CDI significantly decreased upon treatment with immunoglobulin, from 29 to 6 cases (20.6% vs. 6.4%, *p* = 0.003). Therefore, this information may suggest that the immunoglobulin plasma concentration should be measured in the case of CDI or recurrence of this disease in patients after heart transplantation. In the case of hypogammaglobulinemia, intravenous immunoglobulin therapy should be used [52]. In a meta-analysis of data from 30 clinical trials carried out by Paudel et al., the incidence of CDI after heart transplantation was 5.2% [13]. In a retrospective study of 246 heart transplant patients and 8 heart and lung transplant patients, by Bruminhent et al., the incidence of CDI was 8.7%. In heart transplanted patients, CDI was also found as an independent risk factor for mortality (HR 7.66; 95% CI: 3.41–17.21, *p* < 0.0001) [53].

## 7. Hematopoietic Stem Cell Transplantation

*Clostridioides difficile* infection is the leading cause of infectious diarrhea in allogeneic hematopoietic stem cell (HSCT) transplant recipients. After allogeneic HSCT, patients seem to be one of the most vulnerable populations, with CDI rates exceeding even about 25% within 100 days after HSCT [54]. Dysbiosis caused by cytotoxic chemotherapy, immunosuppression and antimicrobial exposure is likely to increase the risk of developing CDI in recipients of allo-HSCT compared to the general hospitalized population. Other risk factors in these patients are: older age, prolonged hospitalization, immunosuppressive treatment and low plasma concentration of protective antibodies due to myeloablative conditioning or unreconstructed immunity after HSCT [55]. In addition to the increased risk of CDI, dysbiosis in these patients may also affect the development of acute graft-versus-host disease (aGVHD). The mechanism of the aGVHD is a complex process that may be initiated by damage to the gastrointestinal mucosa by conditioning chemotherapy or radiotherapy. This can lead to pro-inflammatory cytokine secretion by damaged tissues, activation of antigen-presenting cells, proliferation and induction of cell mediators, such as different cytokines or TNF-α, and further tissue damage. Gastrointestinal dysbiosis can also trigger the activation of host antigen-presenting cells, leading to immune dysregulation and the development of aGVHD [56]. In a retrospective cohort study of 656 recipients undergoing allo-HSCT, Jabr et al. showed that 419 (64%) developed aGVHD and 111 (17%) were diagnosed with CDI within the first 100 days after transplantation. It was also found that the occurrence of CDI was significantly associated with the development of aGVHD (HR 1.52; 95% CI: 1.17 to 1.97; *p* = 0.0018) [57]. In another retrospective analysis, Rosignoli et al. analyzed CDI incidence between the start of conditioning and 100 days after HSCT in a group of 481 patients who underwent autologous (220 patients) or allogeneic HSCT (261 patients). CDI incidence was 5.4%, without significant difference between the two types of HSCT [58]. Amberge et al., in a group of 727 patients with acute myeloid leukemia or myelodysplastic syndrome who underwent alloHCT, found CDI in 13% of cases and 14% of patients were identified as asymptomatic carriers of *Clostridioides difficile*. In addition, patients with CDI had a shorter median overall survival of 8 months compared with 25 months in patients without CDI (HR 1.4, *p* = 0.04) [59].

## 8. Conclusions

As seen in the above review, SOT patients and patients after HSCT are particularly at risk of developing CDI. It is associated with risk factors specific to these groups, such as immunosuppressive therapy, and the increased frequency of antibiotic treatment and hospitalization. The diagnostic procedures for CDI in SOT patients are similar to in the general population and based on the current guidelines published by the *Infectious Diseases Society of America* (IDSA) and the *Society for Healthcare Epidemiology of America* (SHEA) in 2017 [14]. In addition, the treatment of CDI in this group of patients does not differ from the general population, which was specified in the above-mentioned guidelines and discussed in the above article. SOT patients and patients after HSCT are more likely to develop relapses and severe forms of CDI, so it is also essential to prevent this infection. Non-specific prophylaxis is recommended to limit the transmission of infection, such as: the use of contact precautions, such as disposable gloves and apron, frequent handwashing with soap and water, environmental disinfection and patient isolation if CDI is found. An important element of prophylaxis may also be the use of some probiotic strains (for example *lactobacillus plantarum* 299v) during and after antibiotic therapy to reduce the occurrence of dysbiosis, which is one of the risk factors for CDI. Other CDI preventive measures, such as bezlotoxumab, prophylactic vancomycin or immunoglobulin supplementation, may also be valuable. CDI in SOT patients and patients after hematopoietic stem cell transplantation may lead to a deterioration in the function of the transplanted organ or complete loss of its function, which is why it is so important to minimize the risk factors for this infection. A summary of the most important information for each transplantation type is shown in Table 1.

## Figures and Tables

**Figure 1 jcm-11-04365-f001:**
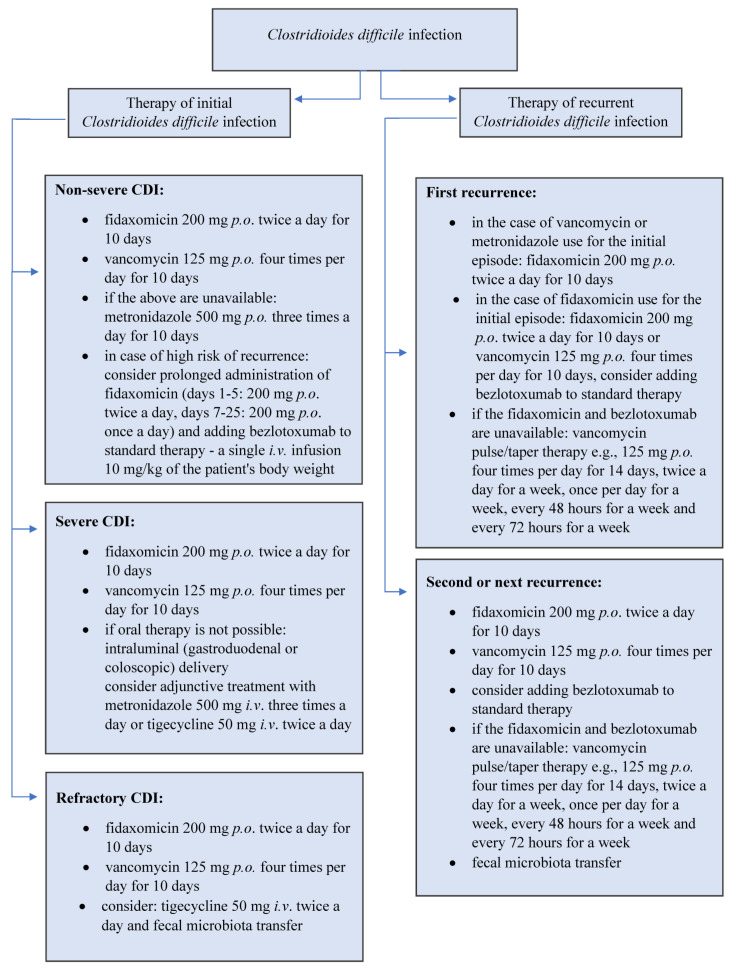
Therapy of *Clostridioides difficile* infection (*i.v*.–intravenous, *p.o*.–*per os*) [22].

**Table 1 jcm-11-04365-t001:** Summary of the most important information concerning CDI in patients after solid-organ transplantation and hematopoietic stem cell transplantation.

Transplanted Organ	Summary of the Most Important Information
**Liver**	incidence of CDI is 9.1% [13] patients with higher MELD scores before liver transplantation had a higher risk of CDI (mean, 24.4 vs. 19.8; *p =* 0.04) [38] CDI was more common in patients with nonalcoholic steatohepatitis 18% vs. 5%, *p =* 0.031) and coinfection with hepatitis C virus and human immunodeficiency virus (11% vs. 1%, *p =* 0.021) [38]
**Kidney**	incidence of CDI is 4.7% [13] disturbances in the composition of the intestinal microbiota-reduced diversity of the microbiome in patients after kidney transplantation compared to the controls (Shannon diversity index, 3.4 vs. 3.7, *p* < 0.001) [44]
**Lung(s)**	incidence of CDI is 10.8% [13] patients with CDI have a higher risk of death (HR 1.61; 95% CI: 1.02–2.52), mainly when CDI occurs in the first 6 months after transplantation (HR 1.96; 95% CI: 1.14–3.36) [48]
**Heart**	incidence of CDI is 5.2% [13] highest incidence of CDI occurs approximately 1 month after transplantation [51] one of the CDI risk factors in these patients is severe hypogammaglobulinemia [49]
**Hematopoietic stem cell**	most vulnerable populations, with CDI rates exceeding even about 25% within 100 days after hematopoietic stem cell transplantation [54] patients with CDI had a shorter median overall survival of 8 months compared with 25 months in patients without CDI (HR 1.4, *p =* 0.04) [59] dysbiosis in these patients may affect the development of acute graft-versus-host disease (aGVHD) [56]

## Data Availability

Not applicable.

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
