# Peer review of "Clostridioides difficile Infection in Patients after Organ Transplantation—A Narrative Overview"

_jcm, 2022, doi:10.3390/jcm11154365_

Round 1

Reviewer 1 Report

The paper is an overall well written review of the literature about post transplant  c difficile infection. 

I have some suggestions to improve the quality of the manuscript:

- page 2-3:  lines 94 to 112 are under the paragraph solid organ transplant, however they are about diagnosis and apply to all population, so I would put them under a paragraph titled : diagnosis

- page 3-4: lines 146-215 are under the paragraph solid organ transplant but again they apply to all infected patients. So I would put these lines in a new paragraph on therapy

- Figure 1: I would add Bezlotoxumab in the figure as well as a possibility in case of recurrences 

- line 346 and 376 the abbreviation HSCT could be used

- the new guidelines by the European Society of Clinical Microbiology and infectious disease published in 2021 should be referenced

- the role of pump inhibitors might be hinted

Author Response

We entirely agree with all  Reviewer remarks.

According to the Reviewer’s suggestion we have created new paragraphs entitled: “Diagnosis” and “Therapy”.

Figure 1 has been redrafted based on the new ESCMID guidelines published in 2021 and supplemented, inter alia, with information on bezlotoxumab.

The abbreviation HSCT was used in the text.

Additionally, information about the use of proton pump inhibitors as a risk factor for CDI was also added.

Reviewer 2 Report

With great interest I read the manuscript “Clostridioides difficile infection in patients after organ transplantation - a narrative overview”. The topic is of high interest to the transplant community and a well written review can improve daily clinical routine. Nevertheless, the manuscript in its current version does not provide a substainial help to clinicans in the field. In my opinion, a high quality review should focus on primary literature, cite essentiall publications helping the reader to find milestone papers fast and offer a quick overview of essential evidence and also on areas where evidence is missing.  Reviews citing other reviews are not helpful but lead to dilution of knowledge and loss of time for readers. Some examples of possible improvments are listed below:

Line 29: "microbiota":  the authors switch the wording "human gut microbiota" and "intestinal microbiota" unnessessarily. Furthermore lines 29 to 32 can be erased as the bring no essential information.

Line  64 "Clostridioides difficile infection is one of the most common causes of antibiotic-associated diarrhea" is based on citation 6, from Roselli et al. It is unclear how the authors base this information on the citation. Please explain.

Reference 10:  the sentence "Solid organ transplant (SOT) patients are more prone to CDI compared to the general population. The main risk factors of Clostridioides difficile infection in these patients are immunosuppressive therapy and frequent antibiotic therapy used for prophylactic or therapeutic purposes" is based on a reference linked to a polish review, as far as I understand a press release for a public information for a ministry. That is not a primary research sources.

Citation 11: Persistent intestinal colonization by Clostridioides 81 difficile is found in 4-15% of the population, and most people have demonstrated periodic colonization [11] - review citing another review.

- Line 158166: Fidaxomicin is a ...... Moreover, no adverse effect of fidaxomicin on the 162 physiological intestinal microbiota was observed. Its advantage is a lower risk of relapse compared to patients treated with vancomycin (15% vs. 25%) due to the prevention of the formation of spores of Clostridioides difficile. Different treatment algorithms are recommended depending on the severity of CDI or the incidence of relapse, as presented in - this is well written and informative. Nevertheless the collegues cite a review from Nagy et al.: "what do we know about the diagnostics, treatment...." as reference for this information. What I would suggest to cite is either a primary literature (publication showing those results) or a meta analysis showed that effect with a combination of different primary papers. For example: Momani et al.: Fidaxomicin vs Vancomycin for the Treatment of a First Episode of Clostridium Difficile Infection: A Meta-analysis and Systematic Review.

Author Response

We completely agree with all Reviewer’s remarks. According to the Reviewer’s suggestion we made appropriate corrections.

The nomenclature of microbiota was standardized throughout the article: the term "human gut microbiota" has been replaced with "intestinal microbiota".

We corrected the incorrect citation 6 in the text - the information came from article: Nagy, E. What do we know about the diagnostics, treatment and epidemiology of Clostridioides (Clostridium) difficile infection in Europe? J Infect Chemother. 2018, 24(3), 164–170

In citation 10, the source was changed from Martirosian, G.; Hryniewicz, W.; Ozorowski, T. Zakażenia Clostridioides (Clostridium) difficile: epidemiologia, diagnostyka, terapia, profilaktyka. Narodowy Instytut Leków, Warszawa 2018 to Revolinski, S. L.; Munoz-Price, L. S. Clostridium difficile in immunocompromised hosts: a review of epidemiology, risk factors, treatment, and prevention. Clin Infect Dis. 2019, 68(12), 2144-2153.

In citation 11 the original article Crobach, M. J. T.; Vernon, J.J.; Loo, V.G.; et al. Understanding Clostridium difficile colonization. Clin Microbiol Rev. 2018, 31, 1094 is cited in place of the review article citing this article (Chong, P.P.; Koh, A.Y.; The gut microbiota in transplant patients. Blood Rev. 2020, 39, 100614).

Citation 15 on fidaxomicin we cites the original article Louie, T. J.; Miller, M. A.; Mullane, K. M.; et al. Fidaxomicin versus vancomycin for Clostridium difficile infection. N Engl J Med. 2011, 364(5), 422-31 and a suggested meta-analysis Momani L. A. Al; Abughanimeh, O.; Boonpheng, B.; et al. Fidaxomicin vs vancomycin for the treatment of a first episode of Clostridium Difficile infection: a meta-analysis and systematic review. Cureus. 2018, 10 (6), e2778 in place of the review article Nagy, E. What do we know about the diagnostics, treatment and epidemiology of Clostridioides (Clostridium) difficile infection in Europe? J Infect Chemother. 2018, 24(3), 164–170

We modified the bibliography in accordance with the recommendations, supplementing the citations with original research and meta-analyzes and removing most of the review papers.

Due to the large number of cited review articles, some of them were removed in favor of original articles containing the cited information. Below is a list of added and deleted articles.

The bibliography was supplemented with:

Oughton, M. T.; Loo, V. G.; Dendukuri, N.; et al. Hand hygiene with soap and water is superior to alcohol rub and antiseptic wipes for removal of Clostridium difficile. Infect Control Hosp Epidemiol 2009, 30 (10), 939–44.

Anderson, D. J.; Chen, L.F.; Weber, D. J.; et al. Enhanced terminal room disinfection and acquisition and infection caused by multidrug-resistant organisms and Clostridium difficile (the Benefits of Enhanced Terminal Room Disinfection study): a cluster-randomised, multicentre, crossover study. Lancet 2017, 389 (10071), 805–14.

Cao, F.; Chen, C. X.; Wang, M.; et al. Updated meta-analysis of controlled observational studies: proton-pump inhibitors and risk of Clostridium difficile infection. J Hosp Infect. 2018, 98(1), 4-13.

Prehn, J. van; Reigadas, E.; Vogelzang. E. H.; et al. European Society of Clinical Microbiology and Infectious Diseases: 2021 update on the treatment guidance document for Clostridioides difficile infection in adults. Clin Microbiol Infect. 2021, Suppl 2, S1-S21.

Vanja, D.; Girault, G.; Branger, C.; et al. Risk factors for Clostridium difficile infection in a hepatology ward. Infect Control Hosp Epidemiol 2007, 28, 202–204.

Borzio, M.; Salerno, F.; Piantoni, L.; et al. Bacterial infection in patients with advanced cirrhosis: a multicentre prospective study. Dig Liver Dis 2001, 33, 41–48.

Harris, A. C.; Young, R.; Devine, S.; et al. International, multicenter standardization of acute graft-versus-host disease clinical data collection: a report from the Mount Sinai Acute GVHD International Consortium. Biol Blood Marrow Transplant. 2016, 22(1), 4-10.

The following review papers were removed:

Smits, W.K.; Lyras, D.; Lacy, D.B.; et al. Clostridium difficile infection. Nat Rev Dis Primers. 2016, 2, 16020

Collini, P.J.; Bauer, M.; Kuijper, E.; et al. Clostridium difficile infection in HIV-seropositive individuals and transplant recipients. J. Infect. 2012, 64, 131–147

Pouch, S.M.; Friedman-Moraco, R.J. Prevention and treatment of Clostridium difficile-associated diarrhea in solid organ transplant recipients. Infect Dis Clin North Am 2018, 32(3), 733-748.

Poutanen, S.M.; Simor, A.E. Clostridium difficile-associated diarrhea in adults. CMAJ, 2004, 171, 51–58

Giacobbe, D.R.; Dettori, S.; Di Bella, S. Bezlotoxumab for preventing recurrent Clostridioides difficile infection: a narrative review from pathophysiology to clinical studies. Infect Dis Ther. 2020, 9(3), 481–494

Trifan, A.; Stoica, O.; Stanciu, C.; et al. Clostridium difficile infection in patients with liver disease: a review. Eur J Clin Microbiol Infect Dis. 2015, 34(12), 2313-24.

Hendrikx, T.; Schnabl, B. Indoles: metabolites produced by intestinal bacteria capable of controlling liver disease manifestation. J Intern Med 2019, 286(1), 32–40.

Gioco, R.; Corona, D.; Ekser, B. Gastrointestinal complications after kidney transplantation. World J Gastroenterol. 2020, 26(38), 5797-5811.

Wong, D.; Nanda, N. Clostridium difficile disease in solid organ transplant recipients: a recommended treatment paradigm. Curr Opin Organ Transplant. 2020, 25(4), 357-363.

Zeiser, R.; Blazar, B.R. Acute graft-versus-host disease - biologic process, prevention, and therapy. N Engl J Med. 2017, 377, 2167–2179.

Shen, L.; Weber, C.R.; Raleigh, D.R.; et al. Tight junction pore and leak pathways: a dynamic duo. Annu Rev Physiol. 2011, 73, 283–309.

Adamczak, M.; Dudzicz, S.; Więcek, A. Zakażenie Clostridioides difficile - co jest ważne dla lekarza rodzinnego? Forum Medycyny Rodzinnej 2020, 14(5), 234-244.

Round 2

Reviewer 2 Report

I believe that the corrections improved the manuscript